# Toroidal diamond anvil cell for detailed measurements under extreme static pressures

Agnès Dewaele[1], Paul Loubeyre[1], Florent Occelli[1], Olivier Marie[1] & Mohamed Mezouar ID [2]

Over the past 60 years, the diamond anvil cell (DAC) has been developed into a widespread high static pressure device. The adaptation of laboratory and synchrotron analytical techniques to DAC enables a detailed exploration in the 100 GPa range. The strain of the anvils under high load explains the 400 GPa limit of the conventional DAC. Here we show a toroidal shape for a diamond anvil tip that enables to extend the DAC use toward the terapascal pressure range. The toroidal-DAC keeps the assets for a complete, reproducible, and accurate characterization of materials, from solids to gases. Raman signal from the diamond anvil or X-ray signal from the rhenium gasket allow measurement of pressure. Here, the equations of state of gold, aluminum, and argon are measured with X-ray diffraction. The data are compared with recent measurements under similar conditions by two other approaches, the double-stage DAC and the dynamic ramp compression.

[1] CEA, DAM, DIF, 91297 Arpajon, France. [2] ESRF, BP220, 38043 Grenoble Cedex, France. Correspondence and requests for materials should be addressed to A.D. (email: agnes.dewaele@cea.fr) or to P.L. (email: paul.loubeyre@cea.fr)

High-pressure physics is an old field for which the process of scientific discovery is driven by new tools. The diamond anvil cell (DAC) was invented to exploit the exceptional properties of diamond in terms of strength and transparency[1]. This centimeter-sized press squeezes samples between two diamond anvils to reach extreme pressures. The use of the DAC has been continuously improved by various developments such as the use of a gasket[2], the anvils bevel shape[3], or better supporting seats[4]. Yet, the possibility to measure reliably and accurately the properties of matter under ultrahigh static pressure relies on two facts. First, the stress state of matter under high static pressure can be well controlled and a quasi-hydrostatic compression of the sample can even be achieved by using a soft solid pressure-transmitting medium or by laser annealing the sample. Second, the sample is macroscopic, at least a few µm in dimensions, enabling the use of a large portfolio of characterization methods in standard laboratory and in front of synchrotron beamlines. It should be acknowledged that nowadays a sample under a 100 GPa pressure can be characterized in great details. Over the past two decades, with extensive use of third-generation synchrotrons, the harvest of results has been striking, pressure seems to turn simplicity in complexity, for instance in hydrogen[5], and properties of materials have been pushed to new limits such as a record critical superconducting temperature[6] or a record chemical energy density storage in polymeric nitrogen[7]. The standard DAC is also a very useful tool to measure the properties of the components of the Earth's interiors up to the thermodynamic conditions of its center (363 GPa, ~6000 K), helping to refine Earth models[8].

Over the past few years, ab initio calculations have disclosed novel properties of matter in the terapascal (TPa) range, due to pressure-induced interaction between core electrons[9] and to the larger fraction of the volume occupied by core electrons which tends to localize valence electrons within interstitial pockets. Subsequent electride structures have been predicted ubiquitous[10,11]. Experimentally, many elements and compounds have already been turned into metals and eventually superconductors[12], but only a few pressure-induced electride phases have been observed[13]. Essential systems for interiors of the giant planets and fundamental physics, hydrogen, methane, or water[14–16], are expected to become metals much above 400 GPa, which is the pressure limit of the conventional DAC[17]. Recently, innovative experimental schemes have been implemented to explore the science in the TPa pressure range.

The double-stage DAC (ds-DAC) could drastically extend the pressure range in static compression experiments: to 600 GPa first[18], and later to 1065 GPa[9,19]. The ds-DAC contains a secondary micro-anvil on the top of the diamond anvil, half-spherical in shape, and made of nano-polycrystalline diamond. Its efficiency is attributed to the added compression by the secondary anvil as well as possibly to the superior mechanical properties of the nanodiamond used to make it (although this is discussed[20]). However, this breakthrough device could not be reproduced with similar success by other teams who reported a large number of unsuccessful runs due to the sliding of the secondary anvils[20–22]. A chemical vapor deposition growth of the second-stage anvil has been suggested to overcome this issue[23]. Also, the sample configuration is quite different from standard DAC, with thinner samples and a stress distribution more heterogeneous over a micrometer length scale[20,22]. Data on the compression of the heaviest metals, from tantalum to gold (Au), have been collected in ds-DAC[9,18,19].

The dynamic compression of solids was made possible up to the TPa range through shockless compression along a quasi-isentropic path[24]. The capability to perform in situ X-ray diffraction on ramp-compressed solids was then developed, and compression curve and structural changes in solids up to the TPa range are now measured[25]. That has recently been illustrated by the observation of the face-centered cubic–hexagonal close-packed–body-centered cubic (fcc–hcp–bcc) phase transitions in aluminum (Al)[26], which had remained a long-pursued project for the static approach[27].

Here we report a new toroidal design of the diamond anvil tip that enables to significantly extend the pressure limit of the DAC and to reach 600 GPa homogeneously on a 5 µm sample. Noting that the pressure limit of the beveled diamond anvils is not caused by any intrinsic instability of the single-crystal diamond but by the large elastic strain at the tip[17,28] and that the diamond phase of carbon should be stable up to roughly 1 TPa[29], we searched for a design of the diamond anvil that would enable to optimize the anvil strain so as to reach pressures up to the TPa range. The concept and first results have been presented in 2015 during an AIRAPT conference[30], but it took us 2 more years to perform the measurements presented below to show that measurements similar to standard DACs are possible, in terms of precision, reproducibility, sample dimensions, and type of materials, from gas to solids, and with any atomic number (here from $Z = 13–79$). The machining of a toroidal groove was made possible by recent advances in focused ion beam (FIB) machining. The new anvil was designed with a dome shape, trying to compensate the deformation of the diamond tip, called cupping, under very high pressure[28]. The dimensions of the toroidal anvil have been optimized by trial and error as shown below. The toroidal shape of the diamond anvil has some similarities with the double-stage diamond of the ds-DAC. The data quality that can be obtained are illustrated in three systems below, Au, Al, and argon (Ar). The Au equation of state (EoS) data are compared to those obtained with a ds-DAC. The Al EoS is compared to the dynamical compression data. In both cases, greater accuracy is achieved using the toroidal-DAC (t-DAC).

## Results

**Preparation of the toroidal diamond anvils.** The toroidal anvils are made by machining a toroidal groove at the tip of (100)-oriented single crystalline commercial anvils, using an FIB module (see Methods section for technical details). At start, the culet size is 25–30 µm beveled to 350 µm with an angle of 10°. In order to optimize the groove shape, maximum pressure and stress and strain distributions have been measured during loadings using synchrotron X-ray diffraction as detailed below. Several (~6) profiles have been tested by this trial and error procedure, varying the groove depth and extension. The one that reached 603 GPa is presented in Fig. 1a, b. In the five runs we discuss here, two central flat diameters were used, $d_1 = 16 \mu m$ and 25 µm for compressing metals (runs 1–4) and a gas (run 5), respectively (see Table 1 in Methods section). Correspondingly, two extensions $d_2$ of the tore have been used, 61 and 75 µm. The geometry chosen for run 5 aimed at increasing the sample size, in the perspective of loading a gas and of studying weak X-ray scatterers by diffraction. The groove depth was between 2.2 and 3 µm to produce a planar disk around the central dome (see Fig. 1d). That is an important feature of the toroidal shape that prevents the gasket from out-flowing. Our measurements show that this pit depth is sufficient to fundamentally modify the pressure distribution near the diamond tip, and to increase the pressure reached by a DAC by a factor of at least 1.5.

These anvils can be classically mounted and aligned in a standard DAC. In the present case, we used a membrane DAC which enables a very fine control of the force applied on the piston. A smooth and slow increase of pressure is needed to allow the relaxation of the gasket and diamond tip under high load.

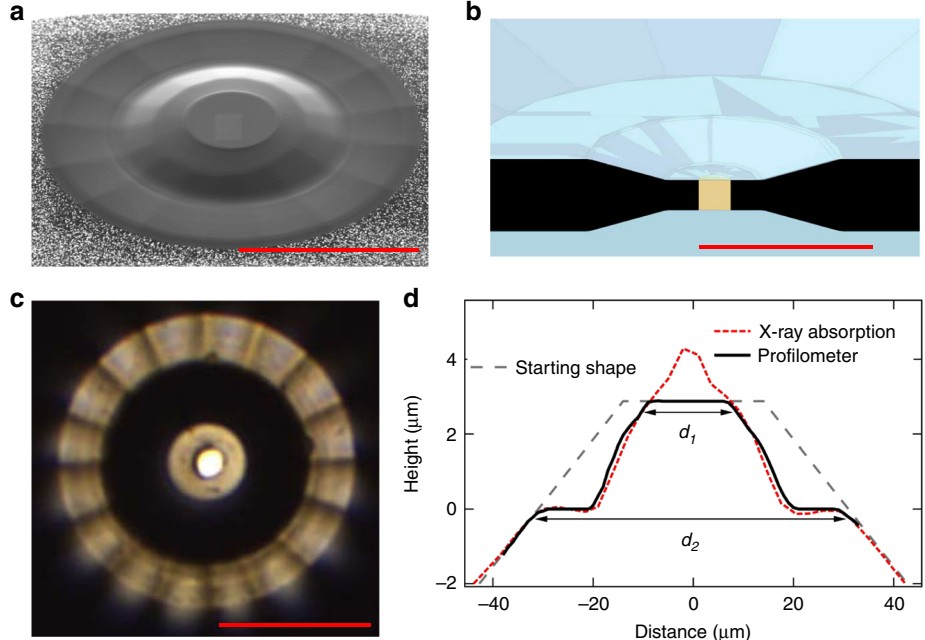

**Fig. 1** Geometry of the optimized toroidal shape used in runs 1–4. **a** Scanning electron micrograph picture. **b** Sketch of the sample chamber in toroidal-DAC. The sample and gasket are represented in yellow and black, respectively. **c** Picture of the aluminum sample in run 4. The sample chamber in the center has been drilled with focused ion beam. The central flat, diameter $d_1 = 16 \, \mu m$, is shiny. The groove has a total extension $d_2 = 60 \, \mu m$ and appears black for the inner part and shiny for the outer (almost horizontal) part. **d** Profile of the diamond tip measured with a profilometer. The raw X-ray absorption profile measured with $\lambda = 0.3738 \, \text{Å}$ (see Methods section) is also plotted. $d_1$ and $d_2$ are the toroidal pit inner and outer diameters. The red scale bars all indicate $30 \, \mu m$

Rhenium (Re) gaskets of 400 μm initial thickness were used and indented at a pressure of 30 GPa. The sample chamber hole was then drilled by FIB machining to achieve a sub-micron centering on the culet and to keep the integrity of the strain-hardened Re (see Fig. 1c and Supplementary Figs. 1 and 2). The sample chamber was loaded with a metal sphere matching its volume or a gas using a high-pressure loading system. Figure 1c illustrates the quality of the sample chamber preparation, with an Al sample of 6 μm diameter and 5 μm thickness at the start.

**Measured stress and strain**. X-ray measurements have been performed on the ID27 beamline of the European Synchrotron radiation Facility. The process of pressure loading was followed by measuring the sample pressure vs. the membrane pressure (hence the force on the piston). It is possible[28,31], with X-ray absorption profiles, to visualize the diamond anvils elastic deformation during the compression (see Methods section). The cartography of the stress at the diamond tip was obtained from a grid of pressure measurements estimated from the Re X-ray pressure scale (see below), with a resolution of 3 μm. The data presented in Fig. 2 have been collected in run 1, with the Au sample geometry as presented in Fig. 1, and are representative of all experimental runs carried out with this geometry (runs 1–4).

An S-shape (three stages) pressure loading behavior is observed in Fig. 2a, similarly to the universal behavior in standard DAC[17], but here the second stage is much more abrupt, yielding to a ≥200 GPa pressure jump. Stage I is associated to the sample assembly compaction with no significant elastic deformation of the anvil tip. Stage II is accompanied by a large elastic strain around the diamond tip, and the central dome is flattening and even cupping on the center. During this stage, the toroidal tip adopts its final 2-μm-deep belt shape around the central flat, which appears to expand up to ~20 μm in diameter when the load

increases. In stage III, the strain around the diamond tip is almost locked (see Fig. 2d), the largest strain being undergone by the anvils bevels up to 200 μm away from the diamond center. The pressure can be further increased up to a point where the anvils elastic strain compensates the bevel initial angle. A direct contact between the anvils induces their breakdown, as in conventional DACs (see Figure S6).

These strains are qualitatively visualized in photographs taken under high load (450 GPa): in Fig. 3b inset, the dark ring is the toroidal groove inner part and is surrounded by a shiny outer part; the central flat appears in light colors, with a 5 μm Au sample at its center. The elastic strain of the diamond anvils completely compensates the first ~4 μm of the pit which appears to be completely flat. The red shade observed at the center of the t-DAC is caused by the onset of an absorption of visible light by the diamond. Above 500 GPa, the red color turns into dark due to the closure of the diamond electronic gap[32]. We estimate that the sample thickness remains ≥1.5 μm, from the compression of the metal volume loaded in the pressure chamber and the sample diameter measured in photographs taken under high load (see Supplementary Figs. 1 and 2); X-ray absorption profiles show that a part (~0.5 μm) of this thickness is ensured by an important cupping of the diamond anvil (see Fig. 2c).

The pressure distribution mapping (Fig. 2c) around the diamond tip has been performed using the Re gasket X-ray powder diffraction (XRD) signal (see Methods section) one step below the maximum pressure reached in run 1. The pressure is homogeneous, within 5%, throughout a 7-μm-wide zone at the center of the anvils. The steep pressure gradients, up to 40 GPa/μm, are located near the edge of the central flat; in contrast, the pressure is relatively homogeneous in the bulk of the tore, attaining approximately one-half of the maximum pressure. Here we use a $2.3 \times 2.6 \, \mu m$ full-width at half-maximum X-ray spot, which has a total extension of ≤6 μm; it thus scans a relatively

**Table 1 Information on the experimental runs**

| Run | $d_1$ (µm) | $d_2$ (µm) | sample | Raman | $P_{max}$ (GPa) |
|-----|-----------|-----------|--------|-------|-----------------|
| 1 | 16 | 60 | Au | N | 603 |
| 2 | 16 | 60 | Au+KCl | N | 443 |
| 3 | 16 | 60 | Al+Au | Y | 317 |
| 4 | 16 | 60 | Al | N | 368 |
| 5 | 25 | 75 | Ar | Y | 429 |

$d_1$ and $d_2$ are defined in Fig. 1. Rhenium gaskets were used for all runs. $P_{max}$ indicates the maximum pressure reached according to Re EoS[33]

compress the sample with no gasket, resulting in very steep pressure gradients even at the center[20–22]; this induces a contamination of the sample XRD signal by a low pressure signal. Dubrovinskaia et al.[19] describes one run using a Re gasket in ds-DAC: its intense XRD signal, at a pressure ~7 times lower than the sample pressure, is collected at the center, also pointing to very important pressure gradients. In our tests, a t-DAC design with a deeper groove has been rejected because of the high-pressure gradients produced at the center (see Supplementary Figs. 3 and 4).

**Pressure measurement**. There is no absolute pressure scale for determining pressure in the range covered here. The use of an X-ray pressure calibrant is at present the only method to estimate the pressure in DACs in the 500 GPa pressure range[9]: the pressure dependence of the calibrant lattice parameters (in other words, the EoS) is assumed on the basis of previous studies; the measurement of these lattice parameters yields the pressure. In our experiments, the calibrant is the Re gasket placed in contact with the sample; this method has been shown to produce reliable results in conventional DACs[33]. The accuracy of the pressure estimate is limited by the accuracy of the X-ray pressure calibrant EoS. Two EoS have been established above 100 GPa for Re: in one study, quasi-hydrostatic compression together with a Au pressure marker[34] constrained the EoS up to 165 GPa and non-hydrostatic compression in ds-DAC up to 608 GPa[18]. A subsequent study[33] claimed that this EoS[18] largely overestimates the pressure (by 40 GPa at 200 GPa, and more than 100 GPa at 400 GPa), a conclusion supported by a recent report[20]. We use the conservative EoS from Anzellini et al.[33], which has been established using measurements performed under quasi-hydrostatic compression up to 145 GPa and validated against other EoS up to 255 GPa[33].

The first-order Raman band of the diamond anvil tip at the interface with the sample has been developed as an optical pressure determination method and calibrated to 410 GPa in a standard DAC[35]. As shown in the Supplementary material (Supplementary Fig. 5, Supplementary Table 1, and Supplementary Note 1), Raman measurement performed simultaneously with X-ray diffraction in one run showed that the Raman gauge[35] is working with t-DAC and agrees with the Re gauge pressure measurement. Remarkably, the Raman toroidal diamond edge remains clear with no luminescence signal even above 400 GPa, indicating no sign of plastic deformation inside the diamond. This opens the possibility to perform standard Raman spectroscopy measurements with the t-DAC.

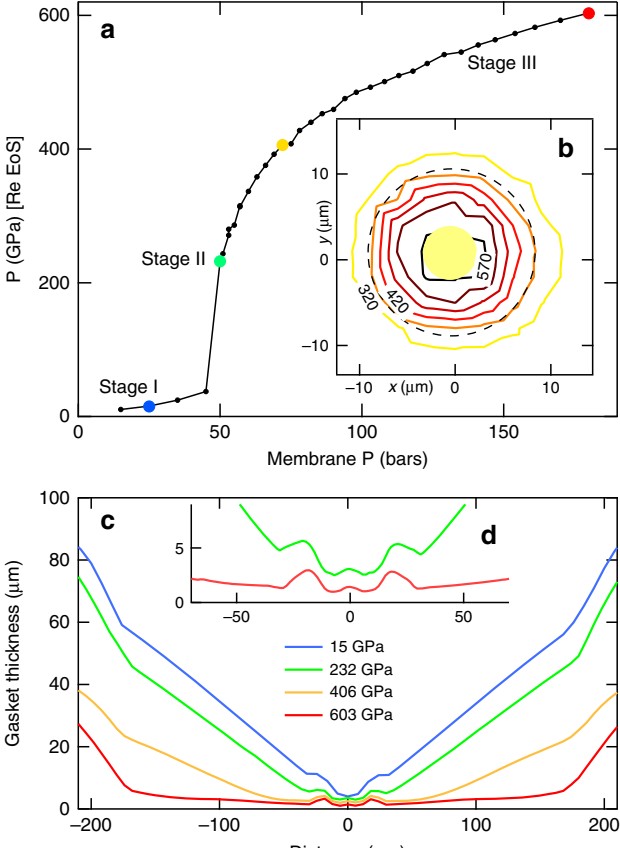

**Fig. 2** Stress and strain measured in run 1. **a** Pressure at the center of the toroidal-DAC (t-DAC) measured using the XRD signal of rhenium gasket and rhenium equation of state[33] vs. membrane pressure. The membrane pressure is proportional to the load applied on the diamond table. The compression stages (I, II, and III) are discussed in the text. **b** Distribution of pressure measured every 3 µm along a 13 × 13 points grid, with the same method. The pressure at the center of the t-DAC is 585 GPa. The dashed line indicates the limits of the central flat and the yellow disc the extension of gold sample. **c** Thickness of the rhenium gasket measured using monochromatic X-ray ($\lambda = 0.3738$ Å) absorption profiles. The profiles have been measured at various pressures indicated by dots with the same color in the sample $P$ vs. membrane $P$ curve in **a**. The red profile corresponds to the data collected just before anvils breakdown. **d** Enlarged view of two profiles: the strain ("cupping") of the central flat can be clearly seen

homogeneous zone, from the pressure point of view, over the whole extend of the Au sample. The obtention of a few µm size sample in diameter and thickness with a weak pressure gradient is very similar to what is achieved in conventional DAC (see the pressure map in Fig. S6). In contrast, the ds-DAC generally

**Equation of state of gold**. A major breakthrough in the development of the DAC measurements has been the possibility to limit the non-hydrostatic stress on the compressed sample by embedding it in a soft "pressure-transmitting medium" (ideally, helium): one refers to such experiments as "quasi-hydrostatic." If such a medium is not used, the yield stress of the sample itself limits the non-hydrostatic stress and subsequent stress gradients. This yield stress $\sigma$ is increasing with pressure and plastic strain[36], which worsens the pressurizing conditions when the DAC load increases, producing experimental artifacts on structure and EoS measurements. In the absence of any soft pressure-transmitting medium, the issue of non-hydrostatic compression has to be considered.

Figure 3a presents the XRD spectra of the Au sample recorded under high load in runs 1 and 2; similar data have been recorded starting around 10 GPa and yield the lattice parameters of Au sample (fcc, space group $Fm\bar{3}m$) and Re gasket hcp structure, space group $P6_3/mmc$) signal under similar pressurizing conditions. Figure 3b plots the EoS of Au up to 603 GPa obtained with

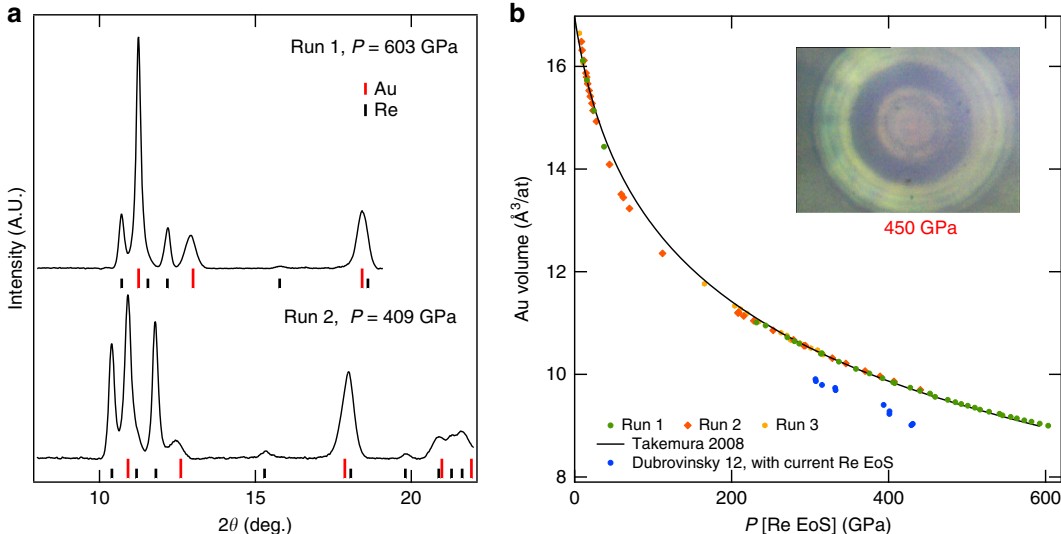

**Fig. 3** Gold equation of state data. **a** XRD spectra collected at the center of the pressure chamber for runs 1 and 2. The exposure time was 20 s. The red and black ticks indicate the positions of the XRD peaks of gold and rhenium, respectively. The (002) peak of gold is slightly shifted from its tickmark under the effect of deviatoric stress (see text). **b** Volume of gold measured up to 603 GPa in runs 1–3. It is based on the (111) diffraction line position, which is the least affected by non-hydrostatic stress[37]. The pressure is estimated using rhenium gasket equation of state [35]. The black continuous line is the quasi-hydrostatic equation of state of gold, measured up to 131 GPa [39] and extrapolated here up to 600 GPa using the Rydberg–Vinet form. The blue dots are joint measurements of Au and Re lattice parameters in the ds-DAC[18], converted to Au equation of state data points similarly to our data points. Inset: photograph of the sample chamber taken at 450 GPa in run 1

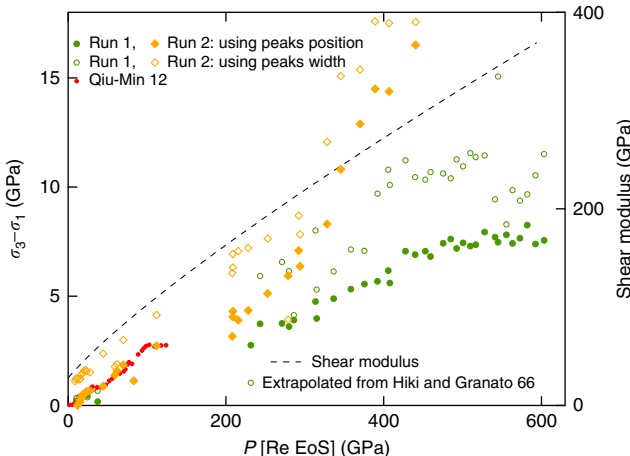

**Fig. 4** Uniaxial stress sustained by gold samples. The uniaxial stress is estimated using two methods:[37,41] comparison between the position of (111) and (200) gold XRD lines (closed symbols), and width of (111) and (200) gold XRD lines (open symbols). The averaged shear modulus of gold[42] is represented as a dashed line for comparison. Recent measurements of gold yield strength up to 127 GPa are also plotted (red dots)[44]

these data (for Au, the (111) XRD peaks which is the least affected by non-hydrostatic compression[37] was used to measure the volume).

The agreement between data collected during runs 1, 2, and 3 attests the reproducible character of our measurements. Above 250 GPa, the EoS agrees well with the extrapolation of an EoS measured under quasi-hydrostatic conditions up to 131 GPa[37] (see Supplementary Table 2 for EoS parameters). This is not fortuitous, as both refs.[33,37] use the same pressure calibrant: the

ruby luminescence gauge calibrated against several X-ray pressure markers[38]. Between 50 and 250 GPa, Au appears to be more compressible here than in the quasi-hydrostatic study[37]. This pressure range corresponds to compression stage II in Fig. 2, when the pressure increases very steeply with increasing load and when a large strain is undergone by the diamond tip. We attribute this apparent higher compressibility to an inhomogeneous stress distribution between the sample and the gasket during this stage, possibly related to an insufficient stabilization time before doing the measurement; at higher pressure, the stress distribution becomes more homogeneous and our data suggest that the pressure exerted on the Re gasket at its center is similar to the pressure in the Au sample.

Au and Re volume data of Dubrovinsky et al.[18] are converted into EoS data point using the same Re pressure scale as here. These data points are plotted as blue dots (EoS) in Fig. 3b and are very far from the EoS of Au; this shows that their joint Re and Au volume measurements, from which the Re EoS was extracted[18], are not compatible with ours. This may be attributed to the heterogeneous stress distribution in ds-DAC, in agreement with pressure maps collected.

We have estimated the non-hydrostatic stress $\sigma_3 - \sigma_1$ (difference between maximal and minimal eigenvalues of the stress tensor) sustained by the Au sample in our experiments, to get a further insight on the reliability of the data collected. Two methods from the literature[37,39–41] have been used: the comparison between diffraction peaks positions (different peaks shift differently with non-hydrostatic stress when the sample is elastically anisotropic); the peaks width evolution (which allows estimating the micro-stress). The equations used are recalled in Supplementary Note 2. For both methods, the single-crystal elastic constants are needed[37]: we extrapolated the low pressure measurements from ref[42]. The evaluated $\sigma_3 - \sigma_1$, expected to be qualitative, is plotted in Fig. 4. This difference is in the same range for the two runs and using the two methods, but the micro-stress calculated using peaks width is systematically higher; this could be related to the pressure heterogeneity (of 20 GPa maximum) in

the zone scanned with X-rays. After a first jump below 100 GPa, $\sigma_3 - \sigma_1$ measured in run 1 increases with pressure approximately proportional to the average shear modulus of Au estimated using low pressure data[42]. This is expected in strength models[43] when the plastic strain is constant. In run 2, the increase is steeper: we explain this by the observed gradual extrusion of the sample to the edge of the central flat, which increased the plastic strain in the sample and thus $\sigma_3 - \sigma_1$ by strain hardening effect. The current estimates are in line with earlier studies[44] in DAC. However, if the same analysis is carried out using ds-DAC data[19], $\sigma_3 - \sigma_1 \simeq -2$ GPa is obtained at 1065 GPa pressure. Both sign and value of stress differ from the observations made here, evidencing a very different stress distribution in the ds-DAC than in the t-DAC and in the conventional DAC.

**High-pressure phases and equation of state of aluminum.** Several theoretical works over the past 30 years[45–47] have predicted a sequence of phase transition fcc–hcp–bcc below 500 GPa in Al ($Z = 13$, $3s^2 3p^1$). The fcc–hcp transition has been observed at 217 GPa[27], but the bcc phase remained outside the reach of standard DACs. Runs 3 and 4 were devoted to the compression of Al, using the same anvils as in runs 1 and 2. In run 3, the Al sample was placed together with one Au grain in the sample chamber. This resulted in a relatively weak XRD signal from Al above ~200 GPa, often overlapping with the Re signal. To get a better constrain on its lattice parameters, we loaded Al alone in the pressure chamber in run 4, which produced an intense XRD pattern (see Fig. 5a). Three phases have been observed: an fcc

$Fm3m$ phase, up to 240 GPa; an hcp $P6_3/mmc$ phase, between 198 and 380 GPa; a phase assigned with a bcc $Im3m$ structure, above 360 GPa. The fcc–hcp coexistence domain is 198–240 GPa; the hcp phase was still observed at 380 GPa. It is interesting to note that in both runs the maximum possible pressure was not reached, probably due to intrinsic defects of the anvils and in one case also to insufficient stabilization time during compression stage II.

The Al EoS measured here (see Fig. 5b) exhibit similar features as Au EoS: during compression stage II, the sample appears more compressible than a quasi-hydrostatically compressed sample[38]. At higher pressure, the two EoS converge. The EoS measured by non-hydrostatic compression in DAC[27] agrees with them, once the pressure which was estimated using a biased EoS of platinum X-ray calibrant is corrected[38] (see Supplementary Table 2). The same correction yields 190–230 GPa for the fcc–hcp coexistence domain, similar to ours.

The transition to a bcc phase has been reported very recently around 321 GPa in a laser ramp-compression experiment[26]. The present data on Al thus offers the possibility of direct comparison between static and dynamic measurements of the phases and EoS of a metal under extreme compression. They overall agree: similar onset pressure for the hcp phase (216 GPa); <5% volume difference for the three phases up to 400 GPa. The bcc phase appears at a slightly lower pressure with dynamic compression than in t-DAC (321 vs. 360 GPa), which can be due to a temperature effect. Indeed, the pressure–temperature path followed during ramp compression is not well controlled, but is always higher in temperature than the principal isentrope[48], which

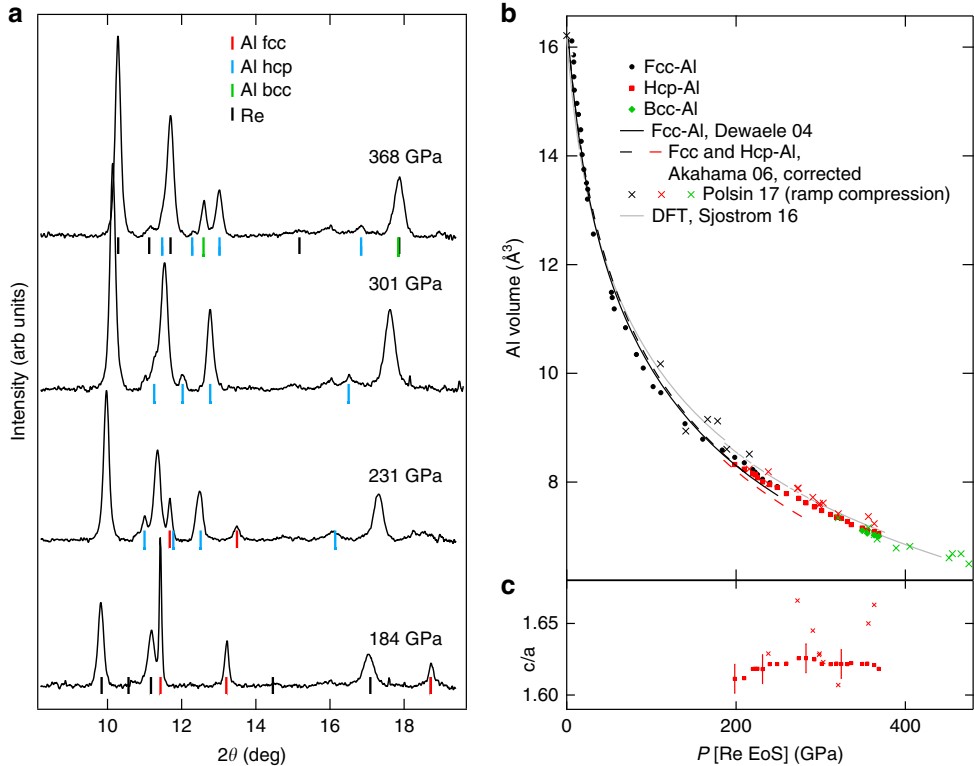

**Fig. 5** Aluminum phases and equation of state data. **a** XRD spectra collected at the center of the pressure chamber for run 4 (aluminum sample in a rhenium gasket). The exposure time was 40 s. The ticks indicate the XRD peaks from the gasket (black) and from the sample (red, blue, and green). Ticks for Re are not plotted for the spectra collected at 231 and 301 GPa. **b** Volume of aluminum measured in run 4. The pressure is estimated using rhenium gasket equation of state[33]. The black line is the quasi-hydrostatic equation of state of fcc aluminum, measured up to 165 GPa[38]. The dashed lines are the equations of state of the fcc and hcp phases measured under non-hydrostatic compression in a standard diamond anvil cell[27], after correction of the pressure metrology (see text). The crosses are volume vs. stress points measured in recent ramp-compression experiments[26], and the gray line the ab initio predicted 0 K equation of state[47]. **c** $c/a$ ratio in hcp-Al measured here and by ramp compression[26]

is estimated to get through 920 K at 364 GPa[26]. Temperature increase usually favors the bcc phase over close-packed phases through the entropy term[49], resulting in negative Clapeyron slopes for hcp → bcc transformations[47]. Other differences cannot be easily explained by the different temperature path: a lower volume measured by ramp compression for the bcc phase, different volume discontinuities for the phase transitions, and different lattice parameters ratio in the hcp phase. The volume discontinuities measured here are −1.0 and −0.8% for the fcc–hcp and hcp–bcc transformations, respectively, in agreement with the DFT predictions[47] and the previous DAC study[27], but lower than the ramp compression measurements (3.2 ± 0.3 and 2.7 ± 0.6%). The $c/a$ ratio for the hcp phase measured is plotted in Fig. 5b: it is centered around 1.62 ± 0.01, in agreement with Akahama et al.[27] (1.618 ± 0.005) but lower than ramp-compression measurement (1.65 ± 0.01[26]). Here, hcp lattice parameters are determined by the measurement of three to four XRD lines (see Supplementary Table 3), vs. three lines for Polsin et al.[26]; however, the parameter $c$ is mostly constrained by the position of the (002) diffraction line, which is weak and asymetric. The error bars on lattice parameters might have been underestimated by Polsin et al[26]. The strength contributions, uncontrollable in dynamic experiments, might also contribute to the volume differences.

**High-pressure equation of state of argon.** The Ar sample loaded in the t-DAC in run 5 was initially formed by a few single crystals, which transformed into a highly textured and strained powder during the compression stage II. The XRD spectrum of this powder shown in Supplementary Fig. 6 exhibits features common in strained fcc crystals[50], with a highly broadened (200) peak and a relatively sharp (111) peak, which corresponds to the densest planes stacking. The (111) peak being the least affected by non-hydrostatic compression[37], it has been used to measure the EoS of Ar up to 248 GPa; above that pressure, this peak overlapped with the most intense Re diffraction peak, hindering the EoS determination. The measured $P−V$ points are plotted in Fig. 6 and agree with the extrapolation of literature data[51,52], collected below 100 GPa (the parameters of the fitted EoS plotted in Fig. 6 are provided in Supplementary Note 3 and Supplementary Table 2). No signal which could be attributed to an hcp modification of Ar could be detected up to 429 GPa. The sample remained perfectly transparent up to the maximum pressure reached (see Supplementary Fig. 2), confirming that the closure of the electronic gap of Ar should occur at higher compression[53].

## Discussion

Over the past 60 years, the developments of the DAC to reach extreme pressures have been made by steps and never at the expense of the quality of the measurements that could be achieved at the time. Here, a further step to reach TPa pressure is demonstrated with the toroidal anvil shape, which was made possible with new FIB machining capability. That modification of the anvil shape is sufficient to increase by a factor of 1.5 the maximum pressure reached compared to a standard design culet of similar diameter. No anvil material with superior mechanical properties is needed[19]. Pressures of 430 and 603 GPa have been obtained using a 25 and 16 μm culets, respectively. Using a 10 μm culet the TPa pressure can thus be envisioned.

The standard of the high-pressure measurements that can be achieved in the DAC is preserved in the t-DAC because the integrity and transparency of the single-crystal diamond anvil, the sample size of roughly 1/3 of culet diameter in dimension, and a homogeneous and well-characterized stress state are preserved.

The toroidal diamond anvils can be mounted in any DAC, as conventional anvils. The handling, alignment procedure, and

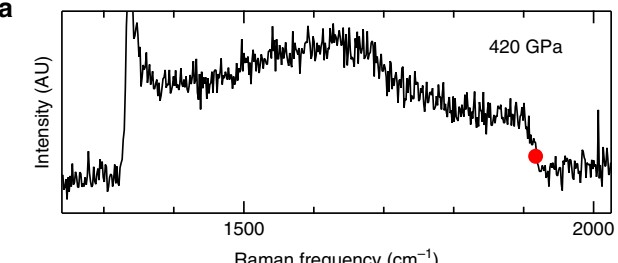

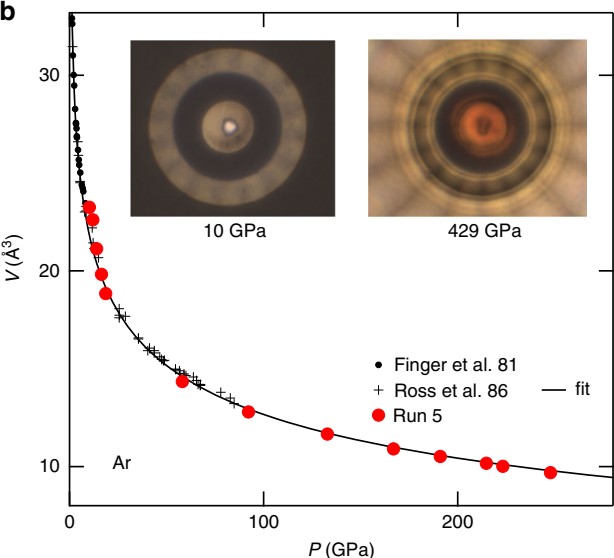

**Fig. 6** Argon equation of state data. **a** Raman spectrum collected at the center of the toroidal-DAC at the end of run 5. The position of the high-frequency edge of the diamond anvil (minimum of the differential spectrum) is indicated by a red disc. It corresponds to a pressure of 420 GPa using the high-pressure calibration[35]. **b** Volume of argon measured in run 5, compared with the equation of state obtained by a fit of lower pressure data points[51,52] (black line). Insets: photographs of the sample. The image at 429 GPa is blurred by birefringence of the diamond anvil under high stress; the red color seen on the diamond tip is attributed to the closure of the diamond band gap[32]

sample loading are identical. The t-DAC can be loaded with any type of samples, from gas to solid, and the μm size homogeneous sample diameter will make possible the study of even the most compressible and weakest X-ray scatterer element, hydrogen. The same pressure metrologies as for conventional DACs in the multi-Mbar regime can be used: X-ray calibrants (such as the Re gasket[33]) as primary gauges and Raman signal of the diamond anvil tip at the sample interface as a secondary gauge[35]. Our measurements show that the Raman edge signal is well preserved up to the maximum pressure, with no parasitic fluorescence signal. This makes possible the use of the t-DAC for laboratory spectroscopy measurements.

The TPa pressure range is the new frontier of extreme condensed matter physics. Two other experimental approaches have already started to produce data in this pressure range. The comparison of the present measurements with the EoS data obtained on Au with ds-DAC and on Al by the ramp compression illustrates the advantage of having kept the standard DAC configuration of the samples. t-DAC can produce reliable static data that will be compared to ramp-compression ones to evidence temperature and time-dependent effects. The data quality will enable researchers to solve some of the current pressure calibration issues[20] above 400

GPa by measuring and comparing the compression curves of more elements.

Finally, we are confident that the possibility of a broad dissemination and use of the t-DAC in high-pressure laboratories and the new possibility offered by the upgrade of synchrotron facilities to better measure μm size samples will help to push the DAC using single-crystal anvils to its final limit that may well be this time the intrinsic stability of the bulk diamond material itself, around 1 TPa.

## Methods

**Focused ion beam machining**. The toroidal shape at the tip of the synthetic single-crystal diamond anvil was made by a FIB delivered by the Field Emission Gun of the FEI Quanta 3D apparatus. The ion beam current used was in between 3 and 5.5 nA under an accelerating voltage of 30 kV. The toroidal pattern was constituted by adjacent concentric crowns with 1 μm width. The duration time of the FIB machining is adapted to give the desired depth of each crown. The total duration of the machining of the toroidal shape of this study was between 2 and 4 h.

**Diamond anvil cell preparation**. High pressure were generated using the LeToullec-type membrane DACs equipped with Boehler-Almax-[4] type seats made of tungsten carbide or polycrystalline diamond. Special care was paid to have a very tight adjustment of the guiding poles. The pressure was very smoothly increased by inflating the membrane with a slow rate of 0.2 bar/min (the conversion between the membrane pressure and the force on the piston is $F(kN) = 0.05 \times P_m$ (bar)).

**X-ray diffraction measurements**. Angular-dispersive XRD (wavelength 0.3738 Å, $2 \times 3 \mu m^2$ spot) was carried out on the ID27 beamline at the ESRF with a bidimensional MAR-CCD detector, with a detector to sample distance calibrated using a reference $CeO_2$ sample. The X-ray beam was focused to a $2.3 \times 2.6 \mu m^2$ spot, and cleaned by two platinum pinholes to remove the wings of the spot. The bidimensional images (examples in Supplementary Fig. 7) were integrated using Fit2D and Dioptas softwares[54,55] and the background was subtracted using Dioptas. The lattice parameters measured for the gasket and the sample are listed in Supplementary Tables 4–7. At each pressure point, the alignment of the X-ray beam with the DAC center was checked, then an XRD exposure was collected at this center (40 s exposure at maximum).

In addition, the DAC was scanned horizontally by ±300 μm, while the signal recorded by a photodiode behind the DAC was recorded, in order to evaluate the strain of the diamond anvils from the X-ray absorption of the Re gasket. It is estimated using X-ray absorption profiles, the initial gasket thickness and XRD intensities at the center of the pressure chamber using the Beer–Lambert law: $ln(I/I_c) = -\mu\rho(e - e_c)$, with $I$ the measured intensity, $e$ the thickness ($I_c$ and $e_c$ being these quantities at the scan center), and $\mu\rho = 0.03876 \mu m^{-1}$ under ambient conditions[56]. The effect of compression was approximately taken into account by increasing this factor $\mu\rho$ on pressure increase, by the compression estimated at the tore pressure.

An XRD mapping was performed on a $13 \times 13$ points grid, every 3 μm, at selected pressure points, to measure the pressure distribution at the diamond tip around the sample chamber, using the Re gasket EoS[33].

**Data availability**. All relevant data are available from the authors.

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

## Acknowledgements
We acknowledge the ESRF for the provision of beamtime under proposals ME-1380, ME-1358, HC-2470, HC-3088, and HC-3682. We thank P. Parisiadis and V. Svitlyk for their experimental help, and D. Polsin for sharing data.

## Author contributions
P.L. designed the project. P.L. and F.O. prepared and loaded the DACs. O.M. performed the FIB machining of the anvils. A.D., F.O., and P.L. conducted the experiments and A.D. analyzed the XRD data. M.M. assisted synchrotron experiments. A.D. and P.L. wrote the manuscript. All authors discussed the results.

## Additional information

**Competing interests:** The authors declare no competing interests.

