## [Peer Review File · Nature Communications]

Reviewers' Comments:

Reviewer #1:

Remarks to the Author:

This is probably the most detailed of a series of recent reports on extending the static pressure range in the diamond cell. The trial and error approach clearly shows the complexity of tuning the pressure distribution over highly stressed anvils and the uncertainty in equations of state demonstrates the notorious problem of pressure estimates in the multi-megabar range. The data collection and interpretation is significantly more solid than in other studies I have seen, and the paper should be published. Still there may be severe problems with the estimates of pressure. The question that the editor has to answer is if this is of broad enough interest for publication in Nature Communications. It is purely a high pressure diamond cell study with, unfortunately, very limited application for even that relatively small community. There is nothing really new to be learned about the equations of state of the materials studied here with regards to pressure accuracy and there is the additional problem of severe deviatoric stresses. The potential for studying metallic hydrogen (where this group plays a leading role) is also questionable due to their own observation of severe changes in the optical properties of diamond that makes optical studies of hydrogen impossible. Moreover, the very steep pressure gradients (40GP/micron) also makes the insertion of electrical leads impracticable.

~~~~~

I have a few more questions directly related to the text:

Abstract: Equations of state are not really measured in a strict sense. This requires measuring both pressure and volume. Here the pressures are obtained from other measurements of volumes of metals without knowing the pressure. It is always an equation with two unknowns. The same applies to the Raman measurements. I think the authors are fully aware of that. These statements have to be reworded.

~~~~~

"It should be acknowledged that nowadays a sample under a 100 GPa pressure can be characterized as finely as under ambient pressure".

I must strongly disagree with this statement. Also 'finely' is an unusual word in this context.

~~~~~

"...nano-polycrystalline diamond. Its efficiency is attributed to the added compression by the secondary anvil as well as to the superior mechanical properties of the material used to make it".

This is not accepted in the HP community

~~~~~

What are "calibrated metal balls"?

~~~~~

"Here we use a  $2.3 \times 2.6 \mu\text{m}$  FWHM X-ray spot, which has a total extension of  $\leq 6 \mu\text{m}$ ; "

From my experience this extension is larger. Real measurements should be presented. But in any case, with a pressure gradient of 40 GPa/ $\mu\text{m}$  described here, the X-rays probe a pressure gradient of 240 GPa. Thus, the pressure gradient contours shown here are most likely flawed.

Why were these experiments done at ID27 and not at a micro-beam beamline?

~~~~~

"In our experiments, the calibrant is the rhenium gasket placed in contact with the sample; this method has been shown to produce reliable results in conventional DACs"

Again, what is reliable. The rhenium equation of state has large uncertainties because of lack of 'reliable' pressure knowledge.

The authors are likely biased by their own (btw excellent) metal x-ray diffraction measurements

where (again) pressures had not been measured independently.

□□□□_____

"We use the conservative EoS from Anzellini et al"

The same argument applies here.

□□□□_____

"We have estimated the non-hydrostatic stress $\sigma_3 - \sigma_1$ (difference between maximal and minimal eigenvalues of the stress tensor) "

The calculations presented here require huge extrapolations of formulations found in the literature and in my opinion the estimate of uncertainties should be more rigorous. I can hardly imagine that the deviatoric stresses presented here are of that low magnitude at 600 GPa. Then the question has to be asked about what is really gained in the present study for a better understanding of equations of state especially when the numbers presented here are based on the measurement of a single diffraction peak.

□□□□_____

"No signal which could be attributed to an hcp modification of argon could be detected up to 429 GPa. "

This is strange because this transition is clearly there as reported in a paper by Errandonea et al. (PRB 2006, not cited here). If the small dimension of the sample is the cause for that, then the present method for studying phase behavior at multi megabar pressures is questionable.

Reviewer #2:

Remarks to the Author:

This paper explores the limits of static compression techniques with diamond anvil cells. A much-discussed technique of using shaped anvils is conducted and claims of pressure as high as 603 GPa are presented. The authors have done careful measurements and analysis, and previous literature is generally well cited. The results could be of broad general interest. As such the paper is suitable for publication in this journal. However, the following should be addressed.

1. Alignment of the second stage is crucial and as is now well recorded, this limits the highest pressure and the stress state of the gasket. A publication by Vohra et al overcomes this issue by using CVD growth using a mask that assures good alignment. This should be referenced [Vohra et al, High Pres Res, 35, 282, 2015]
2. Diamond anvil Raman is a good estimate of the local pressure. Even the best synchrotron microfocussing yields Gaussian profiles which probe a larger sample due to the tails in the distribution (which could also have more intensity than in an ideal Gaussian. Thus, the discussion about the difference in values obtained from peak position and peak width may in fact reflect this rather than an actual variation in stress state as has been presented.
3. The implied claim of pressure precision to three significant in the title is a bit overstated. It should state 600 GPa.

Recommmendation: Publish after revision.

Reviewer #3:

Remarks to the Author:

This is a great paper, which, I believe, represents a real breakthrough in static compression science. While the terapascal limit was broken using a different design of DAC (the ds-DAC) a few years ago, I agree with the authors of this paper that the t-DAC allows the use of traditional cells

and more easily enables the study of liquid and gas samples. The decision of the authors to demonstrate this by studying Ar was a wise one. The sample sizes in the t-DAC are also better matched to synchrotron studies.

This paper should certainly be published without major modifications, but I would ask the authors to consider the following changes and improvements.

The title is a little odd. I would recommend that the authors use a different word to "fine"

Abstract: "400 GPa is accepted as the limit of the DAC," This is only true for the conventional DAC, but the ds-DAC can reach above 1000 GPa. I suggest adding the word "conventional".

P2: "It should be acknowledged that nowadays a sample under a 100 GPa pressure can be characterized as finely as under ambient pressure." I think this is too general. While true for one or two techniques, it is not true in general.

P5: What is a "calibrated metal ball"

P6: "In phase III, the strain around the diamond tip is almost locked (see Fig. 2d), the largest strain being undergone by the anvils bevels up to 200 μm away from the diamond center." It is not clear what the authors mean here. Firstly, by "Fig. 2d" do they mean the inset to Fig. 2c? If so, then this should be labelled panel (d). Secondly, what do they mean by the strain being locked? And finally, what specifically should I be looking at in Fig. 2c (?) that shows me that the "largest strain being undergone by the anvils bevels".

P7: Not sure "pollution" is the best word to use here. Perhaps use "contamination" instead?

Figures

Figure 1: It would help the reader if the original outline of the diamond, prior to using the FIB, were shown in Fig. 1(d) so that they can see exactly what changes have been made to the anvil.

Figure 2: Label to inset to Fig2(c) as Fig2(d) to agree with main text

Figure 3: It is noticeable that the (200) peaks from Au in (a) are displaced from their tickmarks. I understand that this is because the (111) was used to determine the lattice parameter, but the authors may still want to add a point of clarity.

Figure 3: How well does a Vinet or Holzappel-AP2 EoS fit the Au data to ~ 600 GPa. What are the new values of K and K'?

Figure 5: I'd like to see error bars in the inset to (b). It would also be good to see the Polsin c/a ratios in the same plot.

Figure 6: It would be good to know what explicit criterion the authors are using to pick the position of the Raman edge.

I applaud the authors for including tables of d-spacings and lattice parameters etc in the Supplementary material. These type of data are extremely useful to other researchers, and will no doubt be used to try and understand why different groups are obtaining equations of state from their data.

However, the number of decimal places (dp) quoted for different materials at different pressures changes, with no clear explanation as to why. d-spacings are measured to 3 dp in Table S1; 4 dp in

Table SIII; 2, 3 and 4 dp in Table SIV; and 1, 2, 3 and 4 dp in Table SV and SVI. Perhaps there are missing significant 0s in some numbers, or there was varying uncertainty in the measurements from pressure to pressure. But as I believe these tables will be used by other authors, perhaps for constructing new EoSs, I would urge the authors to make them as high-precision as possible.

Response to Reviewers:

Reviewer #1 (Remarks to the Author):

This is probably the most detailed of a series of recent reports on extending the static pressure range in the diamond cell. The trial and error approach clearly shows the complexity of tuning the pressure distribution over highly stressed anvils and the uncertainty in equations of state demonstrates the notorious problem of pressure estimates in the multi-megabar range. The data collection and interpretation is significantly more solid than in other studies I have seen, and the paper should be published. Still there may be severe problems with the estimates of pressure.

Along with his detailed comments on the manuscript (“Equations of state are not really measured in a strict sense. (...) Here the pressures are obtained from other measurements of volumes of metals without knowing the pressure. It is always an equation with two unknowns.” “Again, what is reliable. The rhenium equation of state has large uncertainties because of lack of 'reliable' pressure knowledge.”), the referee seems to question the pressure estimation in this study and more generally, in all DAC studies. We are aware that there is no absolute pressure determination possible in static high pressure devices (DAC and large volume presses). The determination of pressure is based on pre-calibrated gauges such as X-ray gauges (often Au or Pt) from which the pressure is calculated through the equation of state, luminescence gauges (in particular ruby up to 150 GPa) or spectroscopic gauges (in particular, shift of the Raman signal from the diamond tip has been calibrated against X-ray gauges in the multi-Mbar range). All DAC studies use one of these metrologies. An important effort has been and is made by the high pressure community to extend, improve and unify them (this is the current task of an AIRAPT group), and we have taken part to this effort. We thus find arbitrary and unfair to doubt on any report of a high pressure behavior observed in DAC because of the uncertainty on the pressure estimation.

Several metals can be considered as primary gauges as their EoS have been established using reduced shock wave data, for which the pressure is calculated using Rankine-Hugoniot equations. The uncertainty on their calibration has been reduced by cross-check and amounts to 1.5% at 200 GPa under quasi-hydrostatic pressure conditions (Ref 13 in the supplementary materials).

In the present study, we use Re which is a secondary X-ray pressure gauge. The possibility of using rhenium gasket volume was proven valid, when it is measured very close to the sample chamber. The systematic uncertainty is then larger and was estimated to 5% up to 250 GPa in a conventional DAC (Ref 33). Here, with a very similar sample configuration, the Re pressure should be reliable with the same uncertainty. The fact that the very high pressure EoS data points measured for Al, Au and Ar are on the extrapolation of the lower pressure determination EoS, using the physical Rydberg-Vinet form, gives us confidence that the Re pressure scale is indeed operational up to 600 GPa. Furthermore, the diamond anvil Raman edge pressure scale also yields the same pressure up to 420 GPa (the maximum pressure at which such data have been collected). The fact that there is no absolute pressure scale is thus unfair and irrelevant to justify that there may be severe problems in the estimate of pressure. Finally, we give in supplementary materials the value of the Re volume, from which the estimated pressure can be revised in the future if the calibration of the Re pressure gauge is modified.

The question that the editor has to answer is if this is of broad enough interest for publication

in Nature Communications. It is purely a high pressure diamond cell study with, unfortunately, very limited application for even that relatively small community. There is nothing really new to be learned about the equations of state of the materials studied here with regards to pressure accuracy and there is the additional problem of severe deviatoric stresses. The referee expresses here a very subjective view, which contradicts referees #2 and #3 opinion. The possibility to straightforwardly extend the pressure range of current pressure investigations by just changing the shape of the diamond anvil should be of a large interest to many groups in the high pressure community. The exploration of the properties of materials above 300 GPa is a virgin field that just begins to be explored by laser ramp compression, leading to many high profile publications (refs 27, 28, or Smith et al. Nat. Astronomy, 2018). The data are of interest to several communities: planetary physics, condensed matter, materials chemistry, etc. The present t-DAC will democratize such a possibility when coupled with novel large x-ray facilities such as FEL and upgraded synchrotrons. This will then enable a detailed exploration of the properties of matter up to the TPa pressure range. Finally, the EoS data measured here have 5% error bars and are valuable data. The sequence of phase transitions in Al was a long-sought result, recently obtained also by ramp compression (ref 29).

The potential for studying metallic hydrogen (where this group plays a leading role) is also questionable due to their own observation of severe changes in the optical properties of diamond that makes optical studies of hydrogen impossible. Moreover, the very steep pressure gradients (40GP/micron) also makes the insertion of electrical leads impracticable. We disagree. The t-DAC will be perfectly adapted to observe and measure the properties of metallic hydrogen. Pressures in excess of 450 GPa will have to be achieved and so are in the possibility of the t-DAC with few μm samples. The closure of the electronic gap of diamond will not be a problem because the adapted spectroscopic measurements to characterize the properties of hydrogen to its metallic state are infra-red spectroscopic measurements, giving structural transitions and closure of the electronic gap. Probably, the insertion of electrical leads will be more difficult than in conventional DAC but other diagnostics will be implemented to reveal superconductivity such as the use of Mossbauer spectroscopy (as done for H3S) or the promising technique of NV centers implanted in the diamond tip or squids.

I have a few more questions directly related to the text:

Abstract: Equations of state are not really measured in a strict sense. This requires measuring both pressure and volume. Here the pressures are obtained from other measurements of volumes of metals without knowing the pressure. It is always an equation with two unknowns. The same applies to the Raman measurements. I think the authors are fully aware of that. These statements heave to be reworded.

We have modified the abstract by moving the last sentence and by adding Re pressure scale. Modified sentences now read as: 'Raman signal from the diamond anvil or X-ray signal from the rhenium gasket allow measuring pressure. Here, the equations of state of three elements: Au, Al and Ar, are measured with X-ray diffraction using rhenium pressure gauge.'

"It should be acknowledged that nowadays a sample under a 100 GPa pressure can be characterized as finely as under ambient pressure".

I must strongly disagree with this statement. Also 'finely' is an unusual word in this context.

In the text, “fine” has been replaced by “detailed”. Also, the sentence has been replaced by: ‘ It should be acknowledged that nowadays a sample under 100 GPa pressure can be characterized in great details’.

"...nano-polycrystalline diamond. Its efficiency is attributed to the added compression by the secondary anvil as well as to the superior mechanical properties of the material used to make it". This is not accepted in the HP community.

The sentence has been modified to convey this controversy: “Its efficiency is attributed to the added compression by the secondary anvil as well as possibly to the superior mechanical properties of the nanodiamond used to make it (although this is discussed (Sakai 2018)).”

What are "calibrated metal balls"?

To be more explicit that is replaced by: "The sample chamber was loaded with a metal sphere matching its volume".

"Here we use a $2.3 \times 2.6 \mu\text{m}$ FWHM X-ray spot, which has a total extension of $\leq 6 \mu\text{m}$; " From my experience this extension is larger. Real measurements should be presented. But in any case, with a pressure gradient of $40 \text{ GPa}/\mu\text{m}$ described here, the X-rays probe a pressure gradient of 240 GPa . Thus, the pressure gradient contours shown here are most likely flawed. Why were these experiments done at ID27 and not at a micro-beam beamline?

We give the real extension of the FWHM X-ray spot. Special care was paid to focus the beam at best and a $10 \mu\text{m}$ pinhole was used which is not usually used. The steep pressure gradients of $40 \text{ GPa}/\mu\text{m}$ are only located at the edge of the central flat. The pressure is homogeneous over the tip of the anvil (within 20 GPa at 600 GPa , as shown in Fig2b), in a region containing the $5 \mu\text{m}$ sample and the Re gasket part that proved essential to measure reliable EoS data points. The pressure contour is probably inaccurate at the edge of the central flat only.

The absorption scan is a convolution of the extension of the beam with the shape of the diamond. On the scan shown in Fig. 1d, the same extension of the toroidal pit is measured with this scan and with the profilometer, within a few microns: this small difference represents the extension of the X-ray beam.

We did not use the sub-micron beamline of the ESRF because these are not optimized for DAC studies. Structural data would have been degraded (detector sample distance not precisely calibrated). The XYZ stage is not adapted to the movements made in typical DAC experiments and not dimensioned to support the weight of the DAC and the force exerted by a capillary. Such constraints would have been impossible to deal with while performing many trial-and-error experiments, with hundreds to data points collected for each of them.

We agree that when adapted to high pressure studies, such beamlines will be extremely useful to characterize samples compressed in the t-DAC developed here.

"In our experiments, the calibrant is the rhenium gasket placed in contact with the sample; this method has been shown to produce reliable results in conventional DACs"

Again, what is reliable. The rhenium equation of state has large uncertainties because of lack of 'reliable' pressure knowledge.

The authors are likely biased by their own (btw excellent) metal x-ray diffraction measurements where (again) pressures had not been measured independently.

"We use the conservative EoS from Anzellini et al"
The same argument applies here.

This criticism for not using an absolute pressure measurement has been answered above.

"We have estimated the non-hydrostatic stress $\sigma_3 - \sigma_1$ (difference between maximal and minimal eigenvalues of the stress tensor) "

The calculations presented here require huge extrapolations of formulations found in the literature and in my opinion the estimate of uncertainties should be more rigorous. I can hardly imagine that the deviatoric stresses presented here are of that low magnitude at 600 GPa. Then the question has to be asked about what is really gained in the present study for a better understanding of equations of state especially when the numbers presented here are based on the measurement of a single diffraction peak.

The non-hydrostatic stress estimated to be sustained by gold reaches almost 15 GPa at 600 GPa, which is huge for such a soft metal (shear strength of less than 0.1 GPa under ambient conditions). We agree that the non-hydrostatic stress estimation is only qualitative, which is due in part to the uncertainty on the single-crystal elastic constants of Au under extreme compression. This was mentioned in the text p. 9 "The evaluated $\sigma_3 - \sigma_1$, expected to be qualitative,...". We however believe that such analysis provides useful information on the stress state of the sample (as discussed in Ref 41), and a similar analysis on data collected in ds-DAC or ramp compression will certainly be useful.

"No signal which could be attributed to an hcp modification of argon could be detected up to 429 GPa. "

This is strange because this transition is clearly there as reported in a paper by Errandonea et al. (PRB 2006, not cited here). If the small dimension of the sample is the cause for that, then the present method for studying phase behavior at multi megabar pressures is questionable. There is no conflict between our study and Errandonea et al.'s one. The Fcc- hcp modification of Argon could be detected by Errandonea at 49 GPa only after laser heating of the sample. By compressing argon to 86 GPa at room temperature, Ross et al. (J. Chem Phys 1986) did not detect the fcc – hcp transition, which was confirmed by subsequent studies in the same range (see Mao et al. J. Phys. Cond Matt 2006, etc.). In our group we have used argon as pressure medium and did not see any phase change up to 150 GPa at 300K. Either the hcp phase is thermodynamically favored by a temperature increase or/and the fcc-hcp transition is kinetically hindered.

Reviewer #2

This paper explores the limits of static compression techniques with diamond anvil cells. A much-discussed technique of using shaped anvils is conducted and claims of pressure as high as 603 GPa are presented. The authors have done careful measurements and analysis, and previous literature is generally well cited. The results could be of broad general interest. As such the paper is suitable for publication in this journal. However, the following should be addressed.

1. Alignment of the second stage is crucial and as is now well recorded, this limits the highest pressure and the stress state of the gasket. A publication by Vohra et al overcomes this issue

by using CVD growth using a mask that assures good alignment. This should be referenced [Vohra et al, High Pres Res, 35, 282, 2015]

We have incorporated the reference suggested on the method to grow the second anvil directly by CVD to overcome the issue of the alignment of the second stage and modified the text page 3 as: ' A Chemical Vapor Deposition growth of the second stage anvil has been suggested to overcome this issue (Vohra et al HPR 35, 22 (2015)).

2. Diamond anvil Raman is a good estimate of the local pressure. Even the best synchrotron microfocussing yields Gaussian profiles which probe a larger sample due to the tails in the distribution (which could also have more intensity than in an ideal Gaussian. Thus, the discussion about the difference in values obtained from peak position and peak width may in fact reflect this rather than an actual variation in stress state as has been presented.

The relative homogeneity of the measured pressure in the sample chamber does not advocate for this interpretation; however, we agree that this may play a role, and we have modified one sentence p. 8 to: "This difference is in the same range for the two runs and using the two methods, but the micro-stress calculated using peaks width is systematically higher; this could be related to the pressure heterogeneity (of 20 GPa maximum) in the zone scanned with X-rays."

3. The implied claim of pressure precision to three significant in the title is a bit overstated. It should state 600 GPa.

The title has been modified: "603" has been replaced with "600".

Reviewer #3:

This is a great paper, which, I believe, represents a real breakthrough in static compression science. While the terapascal limit was broken using a different design of DAC (the ds-DAC) a few years ago, I agree with the authors of this paper that the t-DAC allows the use of traditional cells and more easily enables the study of liquid and gas samples. The decision of the authors to demonstrate this by studying Ar was a wise one. The sample sizes in the t-DAC are also better matched to synchrotron studies.

This paper should certainly be published without major modifications, but I would ask the authors to consider the following changes and improvements.

The title is a little odd. I would recommend that the authors use a different word to "fine" "Fine" has been replaced by "detailed".

Abstract: "400 GPa is accepted as the limit of the DAC," This is only true for the conventional DAC, but the ds-DAC can reach above 1000 GPa. I suggest adding the word "conventional".

"Conventional" has been added to the text.

P2: "It should be acknowledged that nowadays a sample under a 100 GPa pressure can be characterized as finely as under ambient pressure." I think this is too general. While true for one or two techniques, it is not true in general.

The sentence has been modified to: "It should be acknowledged that nowadays a sample under 100 GPa pressure can be characterized in great details."

P5: What is a "calibrated metal ball"

It means that the size of the metallic sample is chosen so that it exactly fills the sample chamber. To be more explicit the sentence is replaced by: "The sample chamber was loaded with a metal sphere matching its volume".

P6: "In phase III, the strain around the diamond tip is almost locked (see Fig. 2d), the largest strain being undergone by the anvils bevels up to 200 μm away from the diamond center." It is not clear what the authors mean here. Firstly, by "Fig. 2d" do they mean the inset to Fig. 2c? If so, then this should be labelled panel (d). Secondly, what do they mean by the strain being locked? And finally, what specifically should I be looking at in Fig. 2c (?) that shows me that the "largest strain being undergone by the anvils bevels".

Yes, Fig2d refers to the inset in fig2c and the label is now present in the figure (it was indeed missing).

By strain being "locked", we mean the strain of the anvil **tip** does not evolve after the second compression stage (below 230 GPa): fig2d compares tip shapes at 232 GPa and 603 GPa, and they appear to be similar.

In fig2c, you should be looking at the bevel shape (from distance=30 microns to 200 microns). The bevel does not change between 15 and 232 GPa, but its strain is huge between 232 and 603 GPa.

P7: Not sure "pollution" is the best word to use here. Perhaps use "contamination" instead? "Pollution" has been replaced by "contamination".

Figures

Figure 1: It would help the reader if the original outline of the diamond, prior to using the FIB, were shown in Fig. 1(d) so that they can see exactly what changes have been made to the anvil.

Done

Figure 2: Label to inset to Fig2(c) as Fig2(d) to agree with main text

Done

Figure 3: It is noticeable that the (200) peaks from Au in (a) are displaced from their tickmarks. I understand that this is because the (111) was used to determine the lattice parameter, but the authors may still want to add a point of clarity.

A point of clarity is added in the caption: 'the (002) peaks of gold is slightly shifted from its tickmark as the effect of deviatoric stress'

Figure 3: How well does a Vinet or Holzappel-AP2 EoS fit the Au data to ~600 GPa. What are the new values of K and K'?

An additional table (Table SII) listing these parameters has been added p. 8 of the supplementary materials. The EoS parameters compare well with those obtained at lower pressure, as expected from the accordance of P-V points seen in Figs. 3, 5 and 6. The EoS H02 yields slightly lower values of K'0.

Figure 5: I'd like to see error bars in the inset to (b). It would also be good to see the Polsin c/a

ratios in the same plot.

The error bar on c/a is smaller than 0.01, which is now plotted on Fig5b. Polsin et al provided us the c/a values which are now included in the inset of Fig. 5b.

Figure 6: It would be good to know what explicit criterion the authors are using to pick the position of the Raman edge.

The high frequency edge is obtained as the minimum of the differential spectrum, as calibrated by Akahama et al. This is indicated in the figure caption now.

I applaud the authors for including tables of d-spacings and lattice parameters etc in the Supplementary material. These type of data are extremely useful to other researchers, and will no doubt be used to try and understand why different groups are obtaining equations of state from their data.

However, the number of decimal places (dp) quoted for different materials at different pressures changes, with no clear explanation as to why. d-spacings are measured to 3 dp in Table SI; 4 dp in Table SIII; 2, 3 and 4 dp in Table SIV; and 1, 2, 3 and 4 dp in Table SV and SVI. Perhaps there are missing significant 0s in some numbers, or there was varying uncertainty in the measurements from pressure to pressure. But as I believe these tables will be used by other authors, perhaps for constructing new EoSs, I would urge the authors to make them as high-precision as possible.

The relative total uncertainty on interreticular distances measured with XRD is $\sim 5 \times 10^{-4}$, which places the uncertainty on the digit 0.00X or 0.000X for lattice parameters. In general, the tabulated values have the uncertainty on the last digit. One extra digit has been listed when this digit could contain meaningful information (such as for a_{111} and a_{200} , which can be compared in table SIV). The spreadsheet automatically removed "0" at the end of tabulated numbers, which resulted in varying numbers of digits in several columns. We have corrected this in the new version of the supplementary materials.

Reviewers' Comments:

Reviewer #1:

Remarks to the Author:

The authors still have a hard time acknowledging that there is no absolute pressure scale for the pressure range of their study. They should also be honest about the lack of success of the AIRAPT pressure scale task force, which has tried to come to a consensus for decades. We all know the difficulties reducing shock data as we lack knowledge of both the thermodynamics and temperatures. The authors should think about the term "pressure gage". Again, what is measured here is the location of a single diffraction peak from which a volume is calculated and then from that volume a "gage" is created using a different metal as a "gage".

What is remarkable, that at a meeting if somebody claims having reached a tera pascal, somebody else, using a different equation of state for that data set, gets 40 % less. The paper still lacks a more honest analysis of the pressure uncertainties rather than using their own previous data, which in my opinion is not much more than comparing volumes of different metals at similar conditions.

Reviewer #2:

Remarks to the Author:

Editorial Note: This Reviewer provided no further comments for the Authors

Reviewer #3:

Remarks to the Author:

As I said in my first review, this is an excellent paper that heralds a real breakthrough in DAC research. In response to referee 2's comment, I would say that this techniques opens up multi-megabar science to many new researchers, particularly if the FIBed diamonds become commercially available.

The referees have answered all of my questions and queries, and I would then recommend the paper for publication.

I would again commend the authors for all of the additional information inserted into the SuppMat, as it is exactly this kind of detail that will enable other researchers to establish equations of state to ultra-high pressures, and resolve some of the pressure calibration issues raised by Referee 1.

Response to Reviewers:

Reviewer #1:

We have added two sentences to point out the uncertainty in the pressure measurements in this few 100 GPa pressure range.

One sentence at the beginning of the paragraph pressure measurement as ' There is no absolute pressure scale for determining pressure in the range covered here'

Another sentence in the next to last paragraph of the Discussion as ' The quality of the data will enable other researchers to resolve some of the current pressure calibration issues above 400 GPa by measuring and comparing the compression curves of other elements.